# Metastatic Neuroblastoma Presenting as a Submandibular Mass with Mandibular Bone Involvement in a Three-Year-Old Child

**DOI:** 10.3390/ijerph18084157

**Published:** 2021-04-14

**Authors:** Asad Ullah, Atbin Doroodchi, Luis Velasquez Zarate, Samantha N. Mattox, Taylor Sliker, Dorian K. Willhite, Jaffar Khan, Harry C. Owen, Surendra K. Rajpurohit, Nikhil G. Patel, Robyn M. Hatley

**Affiliations:** 1Department of Pathology, Medical College of Georgia at Augusta University, Augusta, GA 30912, USA; aullah@augusta.edu (A.U.); smattox@augusta.edu (S.N.M.); tsliker@augusta.edu (T.S.); dwillhite@augusta.edu (D.K.W.); hcowen95@gmail.com (H.C.O.); 2Department of Surgery, Medical College of Georgia at Augusta University, Augusta, GA 30912, USA; adoroodchi@augusta.edu; 3Department of Pathology, The University of Texas Medical Branch at Galveston, Galveston, TX 77555, USA; luchovlz352@gmail.com; 4Department of Pathology, Indiana University, Bloomington, IN 47405, USA; khanja@iu.edu; 5Georgia Cancer Center, Medical College of Georgia at Augusta University, Augusta, GA 30912, USA; srajpurohit@augusta.edu; 6Department of Pediatric Surgery, Medical College of Georgia at Augusta University, Augusta, GA 30912, USA; rhatley@augusta.edu

**Keywords:** neuroblastoma, ganglioneuroblastoma, pediatric neck mass, malignancy, neoplasm

## Abstract

Although neuroblastoma is one of the most common extra-cranial tumors in the pediatric population, it is rarely seen as a metastasis to the mandibular bone. The following is a case report of a 3-year-old male who initially presented with a submandibular mass that was proven to be a poorly differentiated metastatic neuroblastoma through excisional biopsy. This report is one of the few case reports that demonstrates metastatic submandibular neuroblastoma with mandibular bone involvement in the pediatric population.

## 1. Introduction

Neuroblastoma is the most common solid tumor in the pediatric population and the third most-common pediatric tumor overall [1]. The tumors are derived from neuro-ectoderm cells and can arise from anywhere along the sympathetic chain. About 50–60% of patients with neuroblastoma present with disseminated disease, a finding more commonly seen in patients older than one year old [2]. Metastasis to the head and neck region, especially within the cranium and periorbital regions, and lymph nodes is common; nonetheless, metastasis to the mandibular bone specifically is quite uncommon. Herein we present a male pediatric patient who initially presented with a submandibular mass that was proven to be a metastatic neuroblastoma from the right adrenal gland. Following administration of chemotherapy and surgical excision, the primary tumor was found to contain ganglioneuroblastoma, intermixed subtype, suggesting maturation of the primary tumor, which is typically seen after treatment.

## 2. Case Report

A three-year-old male presented to the emergency department with a one-month history of left submandibular swelling and a one-day history of lower back and left leg pain. The child’s mother stated that he had been holding his left flank and experiencing fever with decreased urine output and constipation. He had been evaluated by his pediatrician for the swelling in his neck and was treated with antibiotics without any signs of improvement. An abdominal ultrasound in the emergency department demonstrated grade I left hydronephrosis. His laboratory results were significant for an elevated erythrocyte sedimentation rate of 120 mm/h (normal 0–19 mm/h), elevated C-reactive protein level at 0.665 mg/dL (normal 0–0.500 mg/dL), and hemoglobin of 9 g/dL (normal 11.0–16.0 g/dL). In addition, an ultrasound of the left submandibular mass revealed an intermediate echogenic mass with very prominent color Doppler flow (Figure 1).

He was subsequently admitted to the hematology/oncology service, and pediatric surgery was consulted for biopsy of the left submandibular mass. On physical examination, a firm left submandibular mass was noted measuring 3 cm × 3 cm. On hospital day 2, an excisional biopsy of the left submandibular mass was performed. The histology revealed small round blue cells with scant pink cytoplasm and focal neuropil structures. The tumor cells were diffusely positive for synaptophysin and ALK-1, negative for S-100, and showed a high Ki-67 proliferation index of 95%. FISH for N-MYC showed no amplification and ALK mutation analysis was negative. Based on the morphologic findings and immunohistochemical profile, the diagnosis of poorly differentiated metastatic neuroblastoma with unfavorable histopathology was made (Figure 2). His urine vanillylmandelic acid and homovanillic acid were 121.7 mg/g (normal <16 mg/g) and 79.3 mg/g (normal <25 mg/g), respectively. MRI performed for staging demonstrated multifocal metastatic lesions involving the cervical, thoracic, and lumbar vertebrae, sacrum, bilateral iliac bones, and the posterior epidural soft tissue component from T6 to T7 causing indentation on the thecal sac without significant cord compression. CT scan demonstrated a 2.9 × 2.2 × 1.4 cm calcified right adrenal mass that was presumed to be the primary site (Figure 3). Bilateral bone marrow biopsies were positive for metastatic neuroblastoma. Furthermore, a nuclear medicine scan demonstrated a lytic lesion at the left mandible (Figure 4). 

Given the degree of metastatic disease and the patient’s age (INRG Stage M; ≥18 months), he was categorized as high-risk with subsequent chemo-port placement on hospital day 3. He began chemotherapy according to COG protocol ANBL0532 and was discharged on hospital day 10 with Sulfamethoxazole/Trimethoprim and polyethylene glycol. After 4 cycles of chemotherapy over 3 months, the patient’s disease was generally stable. He underwent laparoscopic adrenalectomy to remove the primary tumor. Grossly, the lesion showed tan-yellow cut surfaces with focal areas of calcification. Histology revealed extensive Schwannian stromal development occupying more than 50% of the tumor. Multifocal areas with naked neuropil admixed with differentiating neuroblasts and maturing ganglion cells were present. Occasional poorly differentiated neuroblasts and calcified areas were seen. Tumor cells were also positive for synaptophysin, S-100 highlighted the prominent Schwannian stroma, and Ki-67 showed low proliferation index of 6%. The final diagnosis of post chemotherapy ganglioneuroblastoma, intermixed subtype, was made. Following surgery, the patient underwent a 5th cycle of chemotherapy and then completed two peripheral blood stem cell transplants followed by external beam radiation to the primary site at another institution. He then completed 6 cycles of immunotherapy accompanied by isotretinoin according to COG protocol ANBL1531 over a period of 5 months, with continued clinical improvement. 

## 3. Discussion

Neuroblastoma is the most common solid tumor in the pediatric population and the third most common pediatric tumor overall [1]. The vast majority of cases of neuroblastoma are caused by sporadic mutations [3]; however, studies have shown that 1–2% of cases are inherited in an autosomal dominant fashion with the most commonly associated genes being ALK and PHOX2B [4,5]. The tumors are derived from neuro-ectoderm cells and can arise from anywhere along the sympathetic chain [6]. Of primary tumors, 71% are located within the abdominal cavity, 15% in the thorax, and 3% in the head and neck region [7]. However, metastasis tends to involve the bone marrow (70.5%), bone (55.7%), lymph nodes (30.9%), liver (29.6%), intracranial and orbital sites (18.2%), lungs (3.3%), and central nervous system (0.6%) [8]. Approximately 50–60% of patients with neuroblastoma present with disseminated disease, a finding that is more common in patients older than one year old [2]. 

Ganglioneuroblastomas are rare tumors that exist on the same spectrum of peripheral neuroblastic tumors (pNTs) and have intermediate malignant potential according to the International Neuroblastoma Pathology Committee [9]. Tumors require both a mature Schwannian stromal component and a neuroblastic component to be diagnosed as ganglioneuroblastoma [10]. These tumors are further classified as intermixed subtype if the neuroblastic component exists as multiple microscopic foci or as nodular subtype if the neuroblasts form distinct macroscopic nodules [10]. A retrospective review of patients from 1990 to 2014 at a pediatric hospital in Canada identified 17 patients with ganglioneuroblastoma, intermixed subtype, with a median age of 4.7 years at diagnosis [11]. In our case, given the post treatment resection of the adrenal gland, we can comfortably assume that the original lesion was similar in histology to the submandibular mass as it is well documented that these lesions tend to mature after chemotherapy.

It is pertinent to mention the existence of a histologically similar tumor called esthesioneuroblastoma, also known as olfactory neuroblastoma. Such tumor originates from the olfactory neuroepithelium with neuroblastic differentiation. However, these tumors most commonly occur in young adults and show different molecular alterations, that is, MYC amplification. We highlight this differential because both the primary and the most common metastatic sites are in the head and neck region for this specific lesion. 

Although the presentation of neuroblastoma as metastatic disease is quite common, especially in the head and neck region, metastasis to the mandible with both soft tissue and bone involvement at presentation is very rare. The most recent case series of metastatic neuroblastoma to the mandibular bone did not specify the incidence at this specific site, whether as an initial presentation or relapsed disease. They also highlighted that due to its rarity, there are very few reports of this clinical presentation and therefore poor data to accurately estimate the frequency. Thus, reinforcing the idea that metastasis to the mandibular bone is very uncommon, let alone as the presenting lesion [12]. A retrospective review of 279 patients presenting to a tertiary care center in China found that bone marrow is the most common site of metastasis for neuroblastoma, with bone being the second most common site. Metastasis to distant lymph nodes and the mandible only accounted for roughly 6% of cases. Of note, mandibular metastases were more common in ganglioneuroblastoma patients than in neuroblastoma patients [13]. Our review of literature showed 21 cases of mandibular neuroblastoma since 1952; however, in most of these reported cases the patient complained of head and neck symptoms such as tooth mobility, proptosis, or Horner syndrome [14,15,16,17]. Our case is one of the few reported studies in which the patient did not have any of the aforementioned signs or symptoms; rather, the initial presentation consisted of abdominal symptoms and a submandibular mass. 

Surgery is sufficient treatment for most cases of early-stage disease [18]. However, in cases of advanced disease, chemotherapy in conjunction with surgery appears to have a high cure rate for intermediate-risk disease with an overall survival rate of 95% [19,20,21]. Patients with high-risk features (age ≥ 18 months and MYCN amplification) may also have an acceptable clinical outcome; however, more aggressive treatment in the form of combination chemotherapy, radiation therapy, and surgery are typically warranted. Nonetheless, if timely or appropriate management is not rendered accordingly, patients are prone to a poor clinical outcome [22]. 

## 4. Conclusions

An initial presentation of neuroblastoma as a submandibular mass, in our case with metastatic disease involving the mandibular bone and adjacent lymph node, is rare but should be considered as part of the physician’s differential diagnosis along with lymphoma, Langerhans cell histiocytosis, pyogenic lymphadenitis, and cat scratch disease. Metastatic work-up by radiography, bone marrow biopsy, and laboratory investigations (e.g., catecholamine metabolic byproducts) assist with diagnosis, staging, and treatment in cases of metastatic neuroblastoma.

## Figures and Tables

**Figure 1 ijerph-18-04157-f001:**
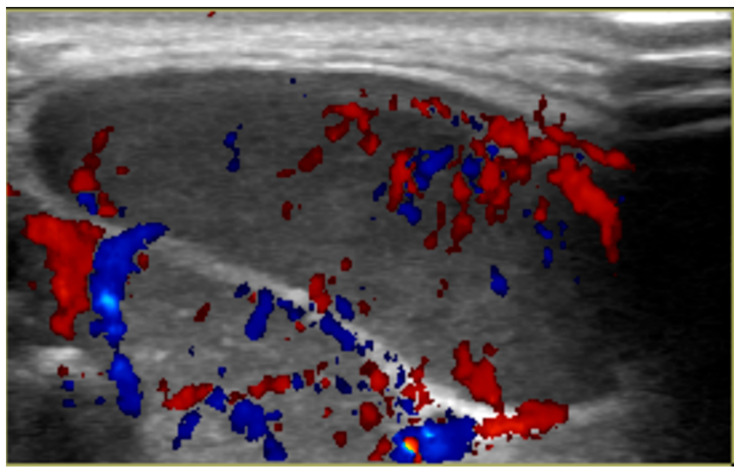
Ultrasound of left submandibular mass demonstrating increased vascularity.

**Figure 2 ijerph-18-04157-f002:**
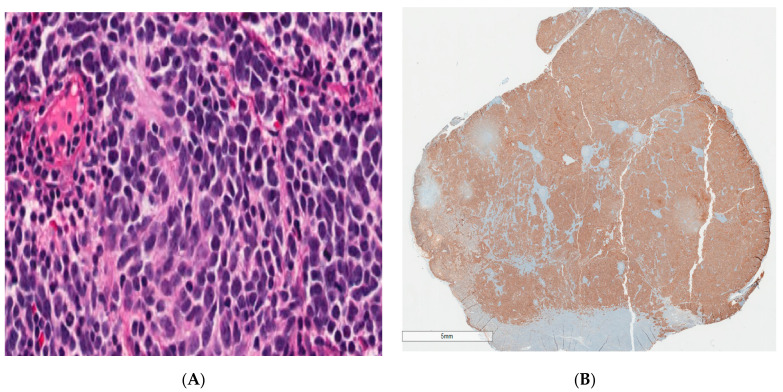
Left submandibular mass biopsy demonstrating malignant cells with neuroblastic differentiation. (**A**): Primitive small round blue cells with high nuclear to cytoplasmic ratio, hyperchromatic nuclei, and scant eosinophilic cytoplasm. Malignant cells are arranged in a syncytial pattern, admixed with intercellular delicate fibrillary processes consistent with neuropil; (**B**): Diffuse and strong expression of synaptophysin immunohistochemical stain.

**Figure 3 ijerph-18-04157-f003:**
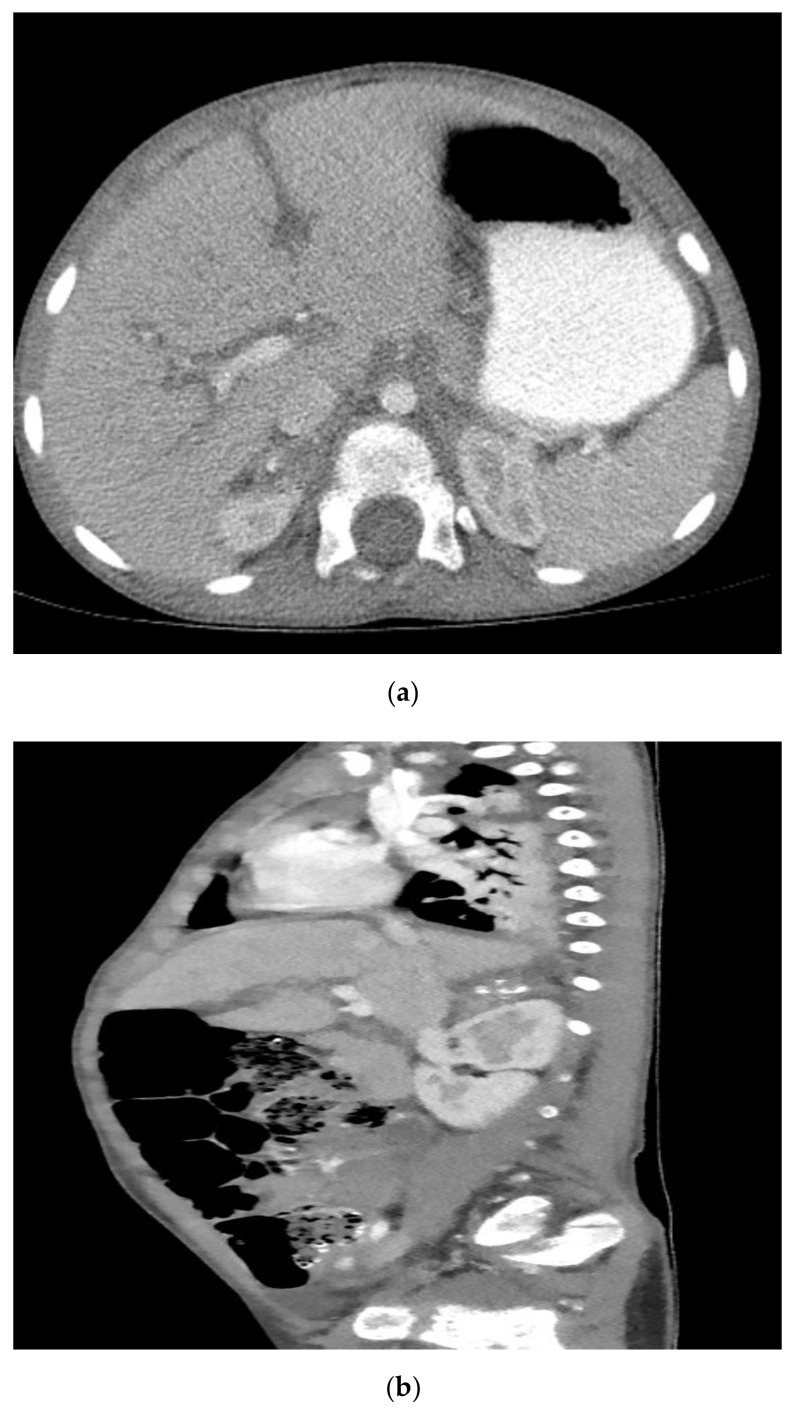
CT scan with contrast in the transverse plane (**a**) and sagittal plane (**b**) demonstrating a calcified right adrenal mass.

**Figure 4 ijerph-18-04157-f004:**
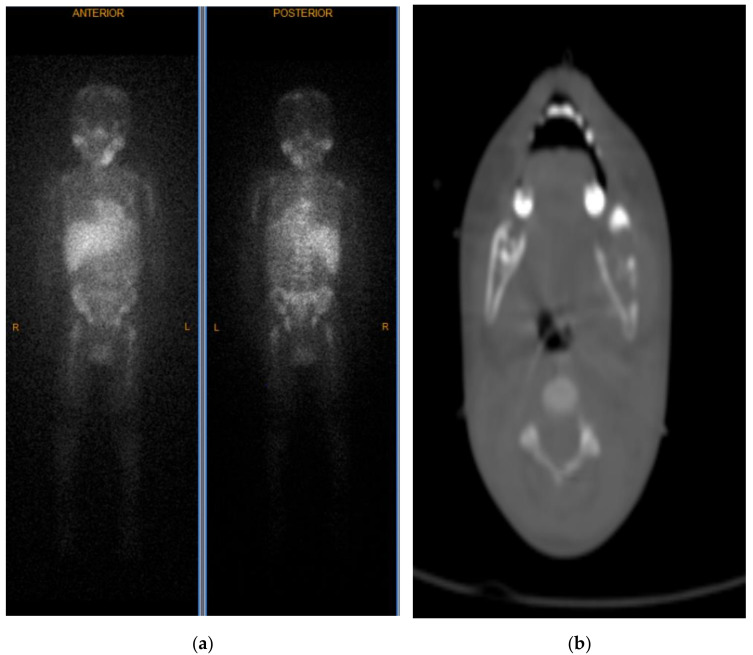
Nuclear medicine scan (MIBG) demonstrating a hyperactive area in the left mandible along with scout CT scan demonstrating a lytic lesion in left mandible in the coronal plane (**a**) and transverse plane (**b**).

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
