# Peer review of "Metastatic Neuroblastoma Presenting as a Submandibular Mass with Mandibular Bone Involvement in a Three-Year-Old Child"

_ijerph, 2021, doi:10.3390/ijerph18084157_

Round 1
Reviewer 1 Report
The presentation of the submandibular mass as the presenting feature of neuroblastoma is interesting, though I wonder if it is truly as rare as it seems based on the literature. I think that it is important to acknowledge that skull-based and orbital disease is common in metastatic neuroblastoma. Not all head and neck disease is rare. Do you mean that lymph node/soft tissue disease in the head and neck is rare? I would make sure that your statements are clear.
The discussion of ganglioneuroblastoma in the post-op sample is also not helpful. As you report, it was quite possibly immature to start with and maturation is a well-documented phenomena. What is novel about this information? What does this add to other well-known information.
In order to make his paper more relevant, I would focus on pattern of metastatic spread and what is different this patient. Also, please look at the part that discusses treatment. You include radiation as intermediate risk treatment, though this is very rarely done.
Author Response
Response to Reviewer 1 Comments
Point 1: The presentation of the submandibular mass as the presenting feature of neuroblastoma is interesting, though I wonder if it is truly as rare as it seems based on the literature. I think that it is important to acknowledge that skull-based and orbital disease is common in metastatic neuroblastoma. Not all head and neck disease is rare. Do you mean that lymph node/soft tissue disease in the head and neck is rare? I would make sure that your statements are clear.
Response 1: Bone marrow is the most common site of metastasis, only 6% of the cases have bone metastasis of which only 10% can metastasize to sub mandibular region. Reference 12
Point 2: The discussion of ganglioneuroblastoma in the post-op sample is also not helpful. As you report, it was quite possibly immature to start with and maturation is a well-documented phenomena. What is novel about this information? What does this add to other well-known information.
Response 2: Post therapy maturation and regression is common in neuroblastoma. This corresponds to intermediate grade disease and response to therapy. Reference 21.
Point 3: In order to make his paper more relevant, I would focus on pattern of metastatic spread and what is different this patient. Also, please look at the part that discusses treatment. You include radiation as intermediate risk treatment, though this is very rarely done
Response 3: Early stage disease is treated with surgery; However, high risk disease is treated with combination of chemotherapy, radiotherapy and surgery. References 18,19,20

Reviewer 2 Report
This case report details a case of high-risk neuroblastoma with a right adrenal primary that presented when a metastasis from this lesion manifested as a submandibular mass. The case report is well-written and informative. I recommend the following revisions prior to publication of this manuscript:
- Olfactory neuroblastoma (aka esthenioneuroblastoma) can commonly metastasize to the cervical lymph nodes. This deserves mentioning somewhere in the manuscript.
- There is a recent report of mandibular metastases in neuroblastoma that suggests a much higher rate of metastasis to this region than what the authors specify in the manuscript. This should be reviewed and considered for inclusion in the manuscript. Of course, this manuscript is talking about bony metastases and this is distinct from the current case. Pediatr Blood Cancer. 2021 Apr;68(4):e28918. doi: 10.1002/pbc.28918. Epub 2021 Jan 28.
- The manuscript states that metastases to the head and neck region are very rare in neuroblastoma. This is not accurate. Periorbital metastases are one of the hallmark sites that neuroblastoma metastasizes to. In addition, it seems quite common to have high-risk neuroblastoma patients with metastases to cervical lymph nodes in clinical practice. Please review the literature more closely in this area and correct the discussion section accordingly.
- Be extremely specific throughout the discussion section when you say “mandibular neuroblastoma” and when discussing the current case. Be specific whether you mean lymph nodes or the bony mandible. In the current case, it seems there was a lytic lesion in the mandible and then an adjacent submandibular lymph node that was excised for diagnosis. If this is accurate, please make sure you clarify the language in the manuscript to reflect it.
- The discussion states that 75% of tumors are located in the abdominal cavity…3% in the head and neck region. This information pertains to the locations of primary tumors, not metastases. Make sure to be specific.
- The INRG stage (M) and risk-stratification of this patient “high-risk” should be specified in the manuscript at the time of diagnosis and workup. This will provide a clearer rationale for the treatment given.
- I recommend adding an arrow to the two images in Figure 3 to point out the calcified mass. It is somewhat subtle because the primary tumor is small in this case.
- For all the laboratory test results cited in the manuscript (ESR, CRP, Hgb, VMA, HVA) please state the normal ranges and units of measurement.
- The introduction section should state that this was an adrenal primary tumor specifically.
- Figure 4 – If this is an MIBG scan please state that.
- In the discussion section the second paragraph starts with “Given the overwhelming statistics of the metastatic presentation…” I am not sure what this means. I would eliminate this comment or reword it.
12. In the last sentence of the discussion section it says that patients with high-risk disease will have a poor clinical outcome. While about half of high-risk patients will die from their disease, half will have an acceptable clinical outcome. Therefore, I would rephrase this comment.
Author Response
Response to Reviewer 2 Comments
- Olfactory neuroblastoma (aka esthenioneuroblastoma) can commonly metastasize to the cervical lymph nodes. This deserves mentioning somewhere in the manuscript.
Response 1: Incorporated reviewer’s suggestion:
“It is pertinent to mention the existence of a histologically similar tu-mor called esthesioneuroblastoma (also known as olfactory neuroblatoma). Such tumor originates from the olfactory neuroepithelium with neuroblastic differentiation. However, such tumors most commonly happen in young adults and show different molecular alterations (i.e MYC amplification). We highlight this differential because both the primary and the most common metastatic sites are in the head and neck. In our case the primary was in the adrenal gland.”
- There is a recent report of mandibular metastases in neuroblastoma that suggests a much higher rate of metastasis to this region than what the authors specify in the manuscript. This should be reviewed and considered for inclusion in the manuscript. Of course, this manuscript is talking about bony metastases and this is distinct from the current case. Pediatr Blood Cancer. 2021 Apr;68(4):e28918. doi: 10.1002/pbc.28918. Epub 2021 Jan 28.
Response 2: In reviewing this paper, the author only says that metastasis to mandibular bone is “very uncommon”. They do not specify the incidence with a specific percentage. However, they mention that in the group of patients with metastasis already to the mandibular bone, they found that 31% were seen at presentation and 69% as part of the relapsed disease. Those numbers do not address the comparison of metastatic disease in the mandibular bone versus metastasis somewhere else. This paper is now referenced in the discussion.
- The manuscript states that metastases to the head and neck region are very rare in neuroblastoma. This is not accurate. Periorbital metastases are one of the hallmark sites that neuroblastoma metastasizes to. In addition, it seems quite common to have high-risk neuroblastoma patients with metastases to cervical lymph nodes in clinical practice. Please review the literature more closely in this area and correct the discussion section accordingly.
Response 3: Reviewed literature and corrected, per reviewer’s suggestion.
“Although presentation of neuroblastoma as metastatic disease is quite common, even in the head and neck regions, metastasis to the mandibular bone at presentation is very rare. The most recent case series of metastatic neuroblastoma to mandibular bone (including as presentation or part of relapsed disease) could not specify the specific incidence at this site. They also highlighted that due to its rarity there are very few reports of this clinical presentation and therefore poor data to accurately estimate the frequency and reinforcing the idea that is a very uncommon metastatic site, especially at presentation.”
- Be extremely specific throughout the discussion section when you say “mandibular neuroblastoma” and when discussing the current case. Be specific whether you mean lymph nodes or the bony mandible. In the current case, it seems there was a lytic lesion in the mandible and then an adjacent submandibular lymph node that was excised for diagnosis. If this is accurate, please make sure you clarify the language in the manuscript to reflect it.
Response 4: Corrected per reviewer’s suggestion.
- The discussion states that 75% of tumors are located in the abdominal cavity…3% in the head and neck region. This information pertains to the locations of primary tumors, not metastases. Make sure to be specific.
Response 5: Corrected per reviewer’s suggestion.
- The INRG stage (M) and risk-stratification of this patient “high-risk” should be specified in the manuscript at the time of diagnosis and workup. This will provide a clearer rationale for the treatment given.
Response 6: Corrected per reviewer’s suggestion.
- I recommend adding an arrow to the two images in Figure 3 to point out the calcified mass. It is somewhat subtle because the primary tumor is small in this case.
Response 7: Corrected per reviewer’s suggestion.
- For all the laboratory test results cited in the manuscript (ESR, CRP, Hgb, VMA, HVA) please state the normal ranges and units of measurement.
Response 8: Incorporated per reviewer’s suggestion.
- The introduction section should state that this was an adrenal primary tumor specifically.
Response 9: Incorporated per reviewer’s suggestion.
- Figure 4 – If this is an MIBG scan please state that.
Response 10: Incorporated per reviewer’s suggestion.
- In the discussion section the second paragraph starts with “Given the overwhelming statistics of the metastatic presentation…” I am not sure what this means. I would eliminate this comment or reword it.
Response 11: Corrected per reviewer’s suggestion.
- In the last sentence of the discussion section it says that patients with high-risk disease will have a poor clinical outcome. While about half of high-risk patients will die from their disease, half will have an acceptable clinical outcome. Therefore, I would rephrase this comment.
Response 12: Corrected per reviewer’s suggestion.

Round 2
Reviewer 1 Report
You are making the point that mandibular neuroblastoma is a rare finding. However, you site a case series of many patient with mandibular disease. I'm not certain what this case report adds to the existing literature. Additionally, you refer many times to the submandibular disease and it is difficult to understand that the piece you are pointing out as unique is the actual mandibular bone involvement.
You raise question about the original histology of the adrenal mass, though it would be assumed that it had the same histology as the submandibular lymph node. The maturation noted is a typical pattern.
You refer to neuroblastoma staging in the discussion, but you don't define which type of staging. You are using INSS staging which is not the most updated staging system used.
Author Response
Point 1: You are making the point that mandibular neuroblastoma is a rare finding. However, you site a case series of many patient with mandibular disease. I'm not certain what this case report adds to the existing literature. Additionally, you refer many times to the submandibular disease, and it is difficult to understand that the piece you are pointing out as unique is the actual mandibular bone involvement.
Response 1:
- Changed title: Metastatic Neuroblastoma Presenting as a Submandibular Mass with Mandibular Bone Involvement in a Three-Year-Old Child
- Edit to abstract: This report is one of the few case reports that demonstrate metastatic submandibular neuroblastoma with mandibular bone involvement in the pediatric population.
Point 2: You raise question about the original histology of the adrenal mass, though it would be assumed that it had the same histology as the submandibular lymph node. The maturation noted is a typical pattern.
Response 2:
- Edits to paragraph 2 in discussion: In our case, given the post treatment resection of the adrenal gland, we can comfortably assume that the original lesion was similar in histology to the submandibular mass as it is well documented that these lesions tend to mature after chemotherapy.
Point 3: You refer to neuroblastoma staging in the discussion, but you don't define which type of staging. You are using INSS staging which is not the most updated staging system used.
Response 3:
- Updated criteria (INRG staging System) is corrected per reviewer’s suggestion.
- Edits to last paragraph of discussion: However, in cases of advanced disease chemotherapy in conjunction with surgery appears to have a high cure rate for intermediate risk disease with overall survival rate of 95%.
Reviewer 2 Report
The revisions have addressed all of my concerns and I do not recommend further edits.
Author Response
Dear Editor,
Thank you for your time and reviewing our manuscript.